# Drug-Related Hypersensitivity Reactions Leading to Emergency Department: Original Data and Systematic Review

**DOI:** 10.3390/jcm11102811

**Published:** 2022-05-16

**Authors:** Silvia Pagani, Niccolò Lombardi, Giada Crescioli, Violetta Giuditta Vighi, Giulia Spada, Paola Andreetta, Annalisa Capuano, Alfredo Vannacci, Mauro Venegoni, Giuseppe Danilo Vighi

**Affiliations:** 1Department of Medicine, ASST Vimercate, 20871 Vimercate, Italy; silvia.pagani@asst-brianza.it (S.P.); paola.andreetta4@gmail.com (P.A.); mauro.venegoni@gmail.com (M.V.); giuseppedanilo.vighi@asst-brianza.it (G.D.V.); 2Section of Pharmacology and Toxicology, Department of Neurosciences, Psychology, Drug Research and Child Health, University of Florence, 50139 Florence, Italy; giada.crescioli@unifi.it (G.C.); alfredo.vannacci@unifi.it (A.V.); 3Tuscan Regional Centre of Pharmacovigilance, 50122 Florence, Italy; 4Hospital Pharmacy, ASST Vimercate, 20871 Vimercate, Italy; giudittavioletta.vighi@asst-brianza.it (V.G.V.); giulia.spada@asst-brianza.it (G.S.); 5Section of Pharmacology “L. Donatelli”, Department of Experimental Medicine, University of Campania “Luigi Vanvitelli”, 80138 Naples, Italy; annalisa.capuano@unicampania.it; 6Campania Regional Centre for Pharmacovigilance and Pharmacoepidemiology, 80138 Naples, Italy; 7Pharmacology Unit, Department of Diagnostics and Public Health, University of Verona, 37100 Verona, Italy

**Keywords:** hypersensitivity, drug allergy, anaphylaxis, emergency department, hospitalization, pharmacovigilance

## Abstract

The aim of the present study is to describe pharmacological characteristics of drug-related allergies and anaphylaxis leading to the emergency department (ED). An 8-year post hoc analysis on the MEREAFaPS Study database was performed (2012–2019). Subjects who experienced drug-related hypersensitivity leading to an ED visit were selected. Logistic regression analyses were used to estimate the reporting odds ratios (RORs) of drug-related allergies and anaphylaxis adjusting for sex, age classes, and ethnicity. In addition, a systematic review of observational studies evaluating drug-related hypersensitivity reactions leading to ED visits in outpatients was performed. Out of 94,073 ED visits, 14.4% cases were drug-related allergies and 0.6% were anaphylaxis. Females accounted for 56%. Multivariate logistic regression showed a higher risk of drug-related allergy among males and all age classes < 65 years, while a higher risk of anaphylaxis was observed for females (ROR 1.20 [1.01–1.42]) and adults (ROR 2.63 [2.21–3.14]). The systematic review included 37 studies. ED visits related to allergy and anaphylaxis ranged from 0.004% to 88%, and drug-related allergies and anaphylaxis ranged from 0.007% to 88%. Both in our analysis and in primary studies, antibacterials, analgesics, and radiocontrast agents were identified as the most common triggers of hypersensitivity.

## 1. Introduction

Drug-related hypersensitivity reactions are a group of adverse drug events (ADEs) that are generally unexpected (Type B reactions—Bizarre) [1] and characterized by symptoms or signs initiated by exposure to a drug at dosages that are usually tolerated [2]. Following the definition proposed by the “International Consensus on Drug Allergy”, hypersensitivity ADEs, which occur in the first few hours after drug administration, are usually characterized by urticaria, angioedema, rhino-conjunctivitis, bronchospasm, and, in the most serious cases, also anaphylaxis [3]. Anaphylaxis is defined as a clinically relevant, generalized, or systemic hypersensitivity ADE that can be life-threatening or fatal [3].

Although the diagnosis of the causative agent can be very difficult, pharmacological treatments are among the leading causes of allergy and anaphylaxis-related deaths in adult individuals and hypersensitivity ADEs remain a serious public health concern both in outpatient and inpatient settings worldwide, due to their high morbidity, mortality, and socioeconomic burden [4].

From an epidemiological point of view, drug-related allergies and anaphylaxis are most frequently triggered by analgesics, antibiotics, biologics, chemotherapeutics, contrast media, nonsteroidal anti-inflammatory drugs (NSAIDs), and proton pump inhibitors, again with age and geographical variations worldwide [4]. Among drug-related anaphylaxis, new triggers have been identified. These include biologics containing α-gal (i.e., cetuximab), small molecules, or novel chemotherapeutics like olaparib [5]. Disinfectants such as chlorhexidine [6], or drug ingredients such as polyethyleneglycol [7], or recently methylcellulose [8], have also been identified as novel substances inducing anaphylaxis. The global incidence of anaphylaxis was estimated to be between 50 and 112 events per 100,000 person per year, with an estimated lifetime prevalence of 0.3–5.1%, depending on the definitions used, study methodology applied, and geographical areas investigated [4,9]. Despite an increasing trend for emergency department (ED) visit and/or hospitalization due to anaphylaxis, its mortality was estimated at 0.05–0.51 per million people/year [4,10].

Taking into consideration the last update of the World Allergy Organization Anaphylaxis Guidance [4], major limitations of epidemiological studies regarding drug-related allergies and anaphylaxis reside in the lack of risk factors/triggers characterization and lack of information on large prospective population-based studies. Moreover, most studies do not differentiate between drug-related hypersensitivity ADEs and other kinds of ADEs, and the diagnosis of hypersensitivity ADEs is mostly based on a suspected clinical history or self-reporting [11].

In this context, pharmacovigilance studies, performed with an “active” approach by trained healthcare professionals, can provide detailed information about hypersensitivity ADEs and their diagnosis, especially when these studies are performed in a hospital setting (i.e., ED) [12,13,14,15,16]. This way, “active” pharmacovigilance studies may represent one of the best epidemiological strategies to fill the above-mentioned major limitations.

The aim of the present study was to describe pharmacological characteristics of drug-related allergies and anaphylaxis leading to ED in Italy, estimating their risk considering subjects’ demographic and clinical characteristics and the most frequently reported suspected drug classes. Furthermore, to complete the evidence obtained with our post hoc analysis, a systematic review of observational studies on drug-related hypersensitivity ADEs leading to ED visit and/or hospitalization was performed.

## 2. Materials and Methods

### 2.1. Post Hoc Analysis

This is an 8-year post hoc analysis performed on the MEREAFaPS Study database [15]. A retrospective observational study was conducted by reviewing all drug-related allergies and anaphylaxis observed in the database between 1 January 2012 and 30 November 2019. Following the Italian Pharmacovigilance legislation, the MEREAFaPS Study, which was conducted in Italy since 2006 and ended in November 2020, collected all ADEs through an ad hoc ADE report form in more than 90 EDs belonging to general hospitals distributed in the national territory in five Italian Regions: Lombardy and Piedmont (north), Tuscany and Emilia-Romagna (centre), and Campania (south). As already stated, the EDs involved in the MEREAFaPS Study allowed good and widespread coverage of the ED Italian population.

For the present analysis, all ADE report forms of drug-related allergy and anaphylaxis were identified with a definite list of MedDRA terms (Appendix A). Subjects who experienced one or more hypersensitivity reactions (allergies and/or anaphylaxis) leading to ED visit were selected, retrieving the following data: demographic information (age, gender, ethnic group); suspected drugs; description of the ADE according to diagnosis and symptoms, codified as detailed by MedDRA dictionary and organized by System Organ Class (SOC) [17]. If present in the MEREAFaPS Study database, the ADE’s outcome “hospitalization”, ADE’s management (i.e., adrenaline use), and the triage colour codes were also recorded. In Italy, the triage codes are divided into four categories and are identified with colours, as follows [18]: (1) “red code” (very critical, life threat, top priority, immediate treatment access); (2) “yellow code” (on average critical, presence of evolutionary risk, possibly life-threatening); (3) “green code” (not very critical, absence of evolutionary risks, deferred services); (4) “white code” (non-critical, non-urgent patients). Anatomical Therapeutic Chemical (ATC) classification system was used to classify suspected drugs [12]. For the pharmacologic subgroup of antibiotics, the 3rd level of ATC class was considered, whereas the 5th level of ATC class was considered for NSAIDs, radiology contrast agents, analgesics, and antineoplastic agents.

Data were summarized using descriptive statistics. Categorical data were reported as frequencies and percentages, while continuous data were reported as median values with the related interquartile ranges (IQRs). Age classes were defined as follows: newborns (from 0 to 1 years); children (from 2 to 11 years); adolescents (from 12 to 17 years); adults (from 18 to 65 years); and elderly (older than 65 years). For each drug class, univariate logistic regression was used to calculate the reporting odds ratios (RORs) of drug-related allergy and anaphylaxis with 95% confidence intervals (CIs) compared to subjects who experienced non-allergy ADEs. Multivariate logistic regression was performed and adjusted for sex, age classes, and ethnicity. All results were statistically significant at *p* < 0.05. Data management and statistical analysis were carried out using STATA 16.1.

The coordinating centre of Lombardy Region (Vimercate, Italy) approved the MEREAFaPS Study in 2006, and the local institutional ethics committee (ASST Monza Ethic Committee) approved the MEREAFaPS Study (Notification number 3724—6 May 2021) according to the legal requirements concerning observational studies [12,13,14,15,16]. Due to the retrospective nature of the present analysis and data anonymization, patient’s consent to participate was not required.

### 2.2. Systematic Review

This is a systematic review conducted following the Preferred Reporting Items for Systematic Reviews and Meta-Analyses [19]. A literature search was performed in PubMed and Embase (last search performed on 25 March 2022). The PubMed search strategy was adapted to the syntax and subject headings of Embase. Records were retrieved on the same day from all sources and the search strategy was updated toward the end of the review, after being validated to ensure it retrieved a high proportion of eligible studies.

We considered for inclusion observational studies, either prospective or retrospective performed in EDs and specifically concerning outpatients, published in English. Randomized clinical trials, reviews and meta-analyses, letters to the editor, case reports, case series, and expert opinions were excluded. We included only articles focusing on drug-related hypersensitivity, anaphylactic reactions, and allergies in outpatients. Moreover, we excluded articles focusing on specific syndromes, such as Steven Johnson and toxic epidermal necrolysis. Two review authors (NL and SP) have independently screened the extracted records and identified the studies for inclusion by screening titles and abstracts yielded by search, eliminating those deemed irrelevant. Full-text articles were retrieved for all references that at least one of two review authors identified for potential inclusion. We selected studies for inclusion based on review of full-text articles. Any discrepancy between the findings of two review authors was resolved through discussion with a third expert (GC).

Data were independently extracted from each article by two authors (SP and GC) using a data collection form. Extracted data included the name of the study authors, year of publication, the country in which participants were recruited, the period of observation, and study design. For each included study, researchers retrieved information regarding: (a) the type of health facility (i.e., community hospitals, tertiary centres, university hospitals); (b) patients’ selection criteria, age, and sex (percentage of females); (c) number of patients analysed in the study, the percentage of ED visits for allergy or anaphylaxis and the number of those related to drugs; (d) number of hospitalization events for drug-related allergy or anaphylaxis; (e) percentage of each causative drug class (if available). Authors of primary studies were contacted to retrieve missing data and/or for additional information. Studies with missing data for two or more of the abovementioned criteria were excluded.

Two review authors (NL and SP) independently assessed the included studies for bias, following the Joanna Briggs Institute Critical Appraisal Checklist for analytical cross-sectional study (last amended in 2017) [20]. For each domain in the tool, a judgment as to the possible risk of bias was made from the information reported in the body of papers. The judgements were made independently by two review authors (NL and SP); disagreements were resolved first by discussion and then by consulting a third author (GC). A graphic representation of potential bias was created using the software RevMan 5.4.1 (https://training.cochrane.org/, accessed on 25 March 2022).

## 3. Results

### 3.1. Post Hoc Analysis

During the study period (2012–2019), out of 94,073 ED visits, 13,532 (14.4%) cases were drug-related allergies, while 548 (0.6%) were anaphylactic events. Females accounted for 56.16%. The mean age was 46 and 55 years for patients who experienced allergies and anaphylaxis, respectively. Overall, 2105 (15.6% out of 13,532) subjects who experienced a drug-related allergy and 371 (67.7% out of 548) subjects who experienced anaphylaxis were hospitalized (Table 1).

As for drug-related allergies, the majority of subjects were female (59.2%), with a mean age of 46.3 ± 22.9 years, and Caucasian (81.72%). The most frequently reported triage colour codes were “green” (46.1%) and “yellow” (17.6%). Multivariate logistic regression showed that females (ROR 0.88 [0.84–0.91]) and subjects aged > 65 years (ROR 0.28 [0.27–0.29]) were at lower risk of drug-related allergies compared to males and other age classes, respectively. On the contrary, a higher risk of drug-related allergy was observed among all age classes < 65 years, in particular in children (age 2–11 years: ROR 2.97 [2.74–3.22]) and in adults (age 18–65 years: ROR 2.48 [2.39–2.58]). “Green” (ROR 1.29 [1.24–1.34]) and “white” (ROR 1.25 [1.15–1.37]) triage colour codes were significantly assigned to subjects experiencing an allergy compared to other triage codes.

Considering anaphylaxis, the majority of subjects were female (52.4%), with a mean age of 55.7 ± 17.7 years, and Caucasian (82.3%). The most frequently reported triage colour codes were “yellow” (33.4%) and “red” (23.5%). Considering anaphylaxis, children (ROR 0.21 [0.07–0.67]) and elderly (ROR 0.45 [0.37–0.53]) were at lower risk of this acute event, while a higher risk was observed for females (ROR 1.20 [1.01–1.42]) and adults (ROR 2.63 [2.21–3.14]). “Red” (ROR 10.68 [8.69–13.13]) and “yellow” (ROR 2.00 [1.68–2.39]) triage colour codes were significantly assigned to subjects experiencing anaphylaxis compared to other triage codes. Subjects who experienced anaphylaxis were statistically associated with a higher risk of hospitalization (ROR 5.62 [4.66–6.79]).

The majority of cases of anaphylaxis were treated with hydration, parenteral steroids, and antihistamines (data not shown). Overall, 58.94% of anaphylaxis (323 out of 548 cases) also reported adrenaline use during ED management (Table 2).

Suspected drug classes and active principles associated with anaphylaxis are reported in Table 3. In our sample, antibacterials for systemic use (ROR 8.75 [7.47–10.25]), NSAIDs (ROR 2.18 [1.71–2.78]), and radiology contrast agents (ROR 11.52 [8.33–15.92]) were significantly associated with a higher risk of anaphylaxis. Among antibacterials, the risk of anaphylaxis was significantly higher for pen, mainly represented by amoxicillin/clavulanate, and cephalosporins, particularly ceftriaxone. Among NSAIDs, the risk of anaphylaxis was significantly higher for dexibuprofen, followed by flurbiprofen, diclofenac, and ketorolac, while among radiology contrast agents, the risk of anaphylaxis was significantly higher for ioprimide, followed by ibitridol, iopamidol, and iomeprol. All other most frequently reported active principles involved in cases of anaphylaxis are depicted in Appendix A.

For all suspected drug classes, the most frequently reported drug-related hypersensitivity reactions affected the skin and subcutaneous tissue (data not shown). In particular, we observed several cases of urticaria, localized or general pruritus, erythema, and rash. These dermatological manifestations showed a different degree of seriousness among patients. Considering the most severe cases, these were represented by systemic reactions, including respiratory distress and anaphylactic shock.

### 3.2. Systematic Review

A total of 832 citations were identified through PubMed and Embase searching. After removing duplicates, 745 citations were screened, of which 657 were excluded as they were deemed irrelevant after title and abstract screening. Eighty-eight citations met inclusion criteria for full-text review (Figure 1).

After full-text review, 37 manuscripts were included in the systematic review (Table 4). Most of the primary studies (57%) were performed in USA and/or Canada [21,22,23,24,25,26,27,28,29,30,31,32,33,34,35,36,37,38,39,40,41], followed by 10 (27%) studies performed in Asia [42,43,44,45,46,47,48,49,50,51], 5 (13%) studies in Europe [52,53,54,55,56], and 1 (3%) study in Australia [57]. Overall, 17 (46%) were multicentre studies, either retrospective or retrospective/prospective [23,24,25,26,28,29,30,31,33,34,35,40,41,42,44,48,57].

Four (10%) studies did not specify whether they were multicentre or single centre studies, while 16 studies were performed in a single ED [21,27,32,39,43,46,47,49,50,51,52,53,54,55,56]. Eight (22%) studies were performed on electronic databases, selecting patients mainly using the International Classification of Diseases (ICD-9 or ICD-10) codes [22,23,24,26,30,36,37,45]. Seven studies (19%) [30,33,34,35,40,56,57] focused on specific drug classes, in particular antibiotics, psychiatric medications, antivirals, and NSAIDs. Twelve studies (32%) included paediatric patients [23,24,26,28,34,35,36,40,41,43,45,55], and females accounted for 34.7% to 73% of selected participants. Overall, female was the most represented sex in the majority of the included studies (24 studies out of 37). The number of included patients varied among the studies, ranging from 21 to 10,848,695. ED visits related to allergy and anaphylaxis accounted for a minimum of 0.004% to a maximum of 88%. According to the study design and selection criteria, drug-related allergies and anaphylaxis ranged from 0.007% to 88%. Eight studies (22%) [21,42,47,48,51,54,55,56] did not report the number of hospitalizations, which varied from 1 to 22,646 patients. The most frequently reported causative drug classes were antibiotics, analgesics and NSAIDs, radiology contrast agents, and anticancer agents.

Quality assessment is depicted in Figure 2. Only 11 studies [23,28,29,31,37,42,45,46,51,55,57] were at low risk of bias for all considered domains. Identification of confounding factors and strategies to deal with them were unclear or at high risk of bias for most of the included studies. In particular, several papers did not report any clear identification of variables for analysis adjustment. Only one study [39] was judged at unclear risk of bias for incomplete description of inclusion criteria, two studies [22,39] for the domain “Exposure measurement”, and only one [40] for “outcome measurement”. Statistical analysis was properly performed in the majority of studies.

## 4. Discussion

The current study summarizes the up-to-date evidence on drug-related allergies and anaphylaxis causing ED visit and hospitalization. Antibacterials for systemic use, NSAIDs, and radiology contrast agents were the most reported drug classes associated with drug-related hypersensitivity reactions.

Although our post hoc analysis showed that female was the sex most represented in allergy events, women seem to be associated with a lower risk of drug-related allergy but a higher risk of anaphylaxis, even if at the limit of statistical significance. Considering drug-related allergy, this evidence is comparable with real-world data coming from observational studies, in which a higher risk of ADEs was usually observed in women [58,59]. Nevertheless, there are no conclusive data that drug-related allergies are more common in females than in males [60]. In general, discrepancy exists regarding the sex difference in allergy caused by different triggers, including pharmacological treatments, with females reporting significantly more allergic reactions in questionnaire studies [61]. Moreover, we also observed that women experienced anaphylaxis more frequently than men. This evidence is comparable with hospital-based studies that suggested a female predominance regarding drug-induced anaphylaxis or a history of immediate penicillin allergy [62,63]. Furthermore, some studies reported that females are twice as likely to have drug-induced anaphylaxis than males [64]. However, the reasons for this sex discrepancy are still incompletely understood [65].

Considering patients’ age, we observed that most drug-related allergies occurred in adults (age 18–65 years). This result can be compared with the evidence published in the literature where it is reported that drug-related allergies typically occur in young and middle-aged adults [66]. In fact, with regards to children, it is well known that they are less likely to be exposed repeatedly to medications, especially due to the absence or lower incidence of comorbidities [59,66]. In our sample, drug-related anaphylaxis was also reported more frequently by adults (age 18–65 years), an age group known to be associated with these serious events, which occur mainly in subjects with a mean age of around 58–60 years [67,68]. Of note, the mean age of patients experiencing an allergy or anaphylaxis in our sample was even lower. Although age is consistently associated with severity of anaphylaxis in many studies [64,69], we did not observe a higher risk of drug-related allergy and anaphylaxis in elderly (>65 years). This may be due to the presence of a relatively low number of elderly subjects who experienced an allergy or anaphylaxis as the cause of ED visits and hospitalization in our sample. Moreover, we hypothesize that the elderly included in our analysis may be represented by subjects mainly exposed to long-term pharmacological treatments (i.e., chronic treatment), which can be considered to be less associated with hypersensitivity reactions.

In Italy, triage assessment is made on a colour code basis, with highest priority given to a red code, followed by yellow, green, and white [70]. Although triage is a very useful tool for prioritizing patients upon their arrival to the ED, limited data relating to the triage assessment colour code for drug-related hypersensitivity reactions are available [71]. Nevertheless, our analysis found that more serious events, in particular anaphylaxis, were correctly coded and were mainly associated with “red” and “yellow” triage codes.

With respect to drug classes most frequently associated with anaphylaxis, striking geographical differences exist and are likely caused by local prescription patterns and are influenced by other less characterized factors, such as genetic differences [64]. Despite this, as highlighted by the results of our systematic review, the findings of the active pharmacovigilance study MEREAFaPS were in line with those of studies already published in literature on this topic. Both in our post hoc analysis and in the studies included in the systematic review, most cases of anaphylaxis were caused by the administration of antibacterials for systemic use, NSAIDs, and radiology contrast agents. In general, the best strategy for drug-related hypersensitivity management is avoidance or discontinuation of the suspected medication. Alternative medications with unrelated chemical structures should be substituted, always considering the presence of drugs cross-reactivity [72]. Antibacterials for systemic use, in particular penicillins, are the most common triggers observed in drug-related allergy, affecting approximately 10% of patients [73]. Another group of antibiotics frequently associated with hypersensitivity are cephalosporins, which are generally causative of maculopapular rashes and drug fever, while urticaria and anaphylaxis are uncommon [74]. Analgesics, in particular NSAIDs, can cause hypersensitivity reactions, including exacerbations of underlying respiratory diseases, urticaria, angioedema, and anaphylaxis [72]. Finally, radiology contrast agents are associated with both allergic and pseudoallergic reactions. The incidence of these reactions, including anaphylaxis, appears to be lower with non-ionic versus ionic agents. Drug-related hypersensitivity reactions to radiology contrast agents can be prevented through pre-treatment regimens with corticosteroids and H1-antihistamines [72]. Intramuscular epinephrine (adrenaline) represents the first-line treatment for anaphylaxis. However, even if its use remains suboptimal [4], in our population epinephrine was used in a relatively high percentage of cases of anaphylaxis. After anaphylaxis occurrence, patients should be referred to a specialist to assess the potential cause and to be educated on prevention of recurrences and self-management. The limited availability of epinephrine auto-injectors remains a major problem worldwide, especially in low- and middle-income countries [75].

### Strengths and Limitations

Our study has some limitations. First, the retrospective nature of the study may have led to an underestimation of allergic and anaphylactic ADEs, because ED physicians could not report the reaction or could not recognize the allergic nature of the reaction. This possibility has been evidenced by several studies, such those published by Sundquist et al. [76] and Martelli et al. [77], who evidenced a reduced capacity to recognize anaphylaxis in ED. Another cause of underestimation of anaphylaxis could be the death of serious cases before their arrival in ED. However, considering that the ADEs were collected through a national active pharmacovigilance initiative, the issue of underreporting, especially for anaphylaxis, can be considered of relatively low relevance. Moreover, we did not evaluate the effect of concomitant medications and comorbidities on hypersensitivity reactions and the seriousness of each drug-related allergy. It should be of great importance to have information on concomitant medications since they may potentiate anaphylaxis symptoms or reduce the efficacy of its treatment [78,79]. Even some comorbidities, such as respiratory and cardiovascular diseases, have been associated with poorer prognosis as they may lead to insufficient compensatory mechanisms to endure anaphylaxis complications [80]. Furthermore, we cannot exclude a partial lack of additional data mainly due to the specific condition of each ED (i.e., lack of time, general conditions of patients, and other emergencies). Our analysis is based on ADE reports that are affected by limits that include inaccurate and incomplete information on patients (i.e., sex, age, ethnicity, triage colour code), also mainly related to lack of clinical data in the ED electronic sources. Finally, we cannot have data on the recurrence of anaphylaxis, which involves 3% of patients within one year [37].

Despite these shortcomings, our study overcomes the limitations listed by the World Allergy Organization Anaphylaxis Guidance [4]. As for lack of large prospective population-based studies, this is the first nationwide multicentre study investigating allergic ADEs as the cause of ED visits in Italy. Our analysis included five Italian Regions located in northern, central, and southern Italy, allowing us to reach an estimated coverage of over 45% of the Italian population (more than 28 million inhabitants) [15]. Few multicentre studies examined such a large population with an active pharmacovigilance approach in ED. Most studies were carried out in single hospitals, involving few EDs, and larger studies have usually been performed on administrative databases (i.e., insurance claims), using ICD-9 and ICD-10 codes [22,23,24,26,30,36,37,45]. The use of ICD codes in observational studies may have been associated with a misclassification of patients, with the possible introduction of a selection bias. Moreover, these codes cover anaphylaxis definition as reported by Regateiro et al. [64] only partially. On the contrary, the use of MedDRA standardized medical terminology in our study allowed us to reach a proper differentiation between drug-related hypersensitivity ADEs and other kinds of ADEs, with a certainty of the diagnosis. In fact, each ADE and diagnosis in pharmacovigilance report forms were identified and then coded by trained monitors, and additional information was requested to ensure the greatest correspondence between what occurred and what was entered in the database, thus minimizing the misclassification and the selection bias in this post hoc analysis. Moreover, while several studies concerning allergies and anaphylaxis did not always report risk factors and triggers [22,24,27,31,32,36,37,43,45], the use of the ATC classification system allowed us to record all active principles causative of the hypersensitivity event with certainty.

## 5. Conclusions

In conclusion, drug-related hypersensitivity reactions represent a relevant clinical issue worldwide. Despite the large number of available marketed medications, pharmacological triggers associated with hypersensitivity are mostly well-known drug classes. Both our post hoc analysis and the systematic review confirmed the association between allergy and anaphylaxis and antibiotics for systemic use, NSAIDs, and radiology contrast agents, especially in women and adults. This information should always be taken into consideration by general practitioners, patients and their caregivers, and ED healthcare professionals, to both minimize the occurrence of drug-related hypersensitivity reactions and improve their management.

## Figures and Tables

**Figure 1 jcm-11-02811-f001:**
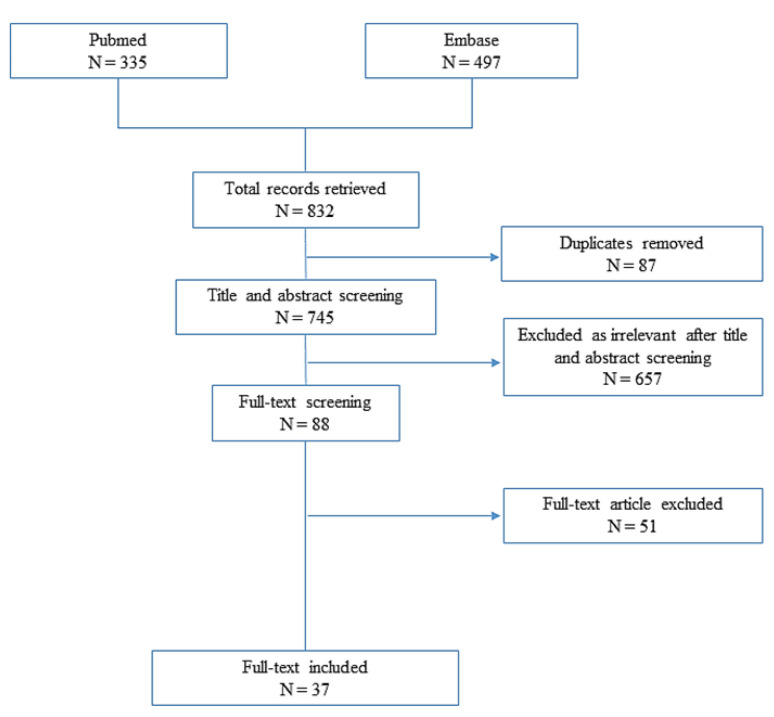
Flowchart depicting article selection.

**Figure 2 jcm-11-02811-f002:**
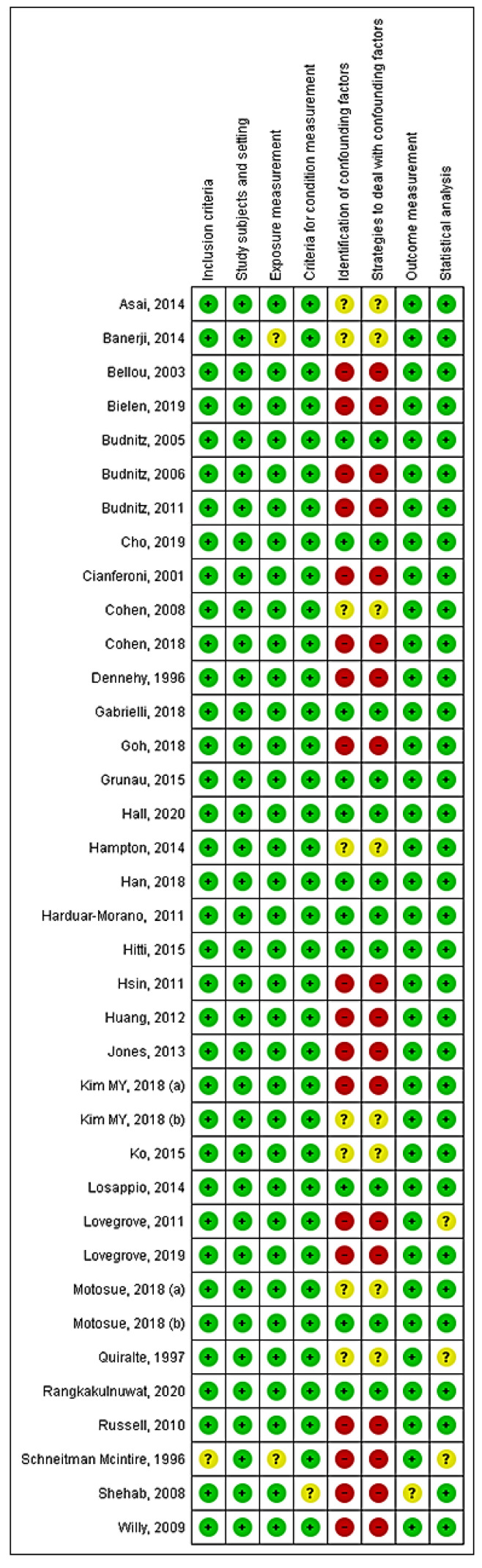
Risk of bias graph assessment, performed according to Cochrane Collaboration’s risk of bias tool [21,22,23,24,25,26,27,28,29,30,31,32,33,34,35,36,37,38,39,40,41,42,43,44,45,46,47,48,49,50,51,52,53,54,55,56,57].

**Table 1 jcm-11-02811-t001:** Risk of drug-related allergy and anaphylaxis by patients’ characteristics.

		Overall N = 94,073 (%)	Non-Allergy N = 79,993 (%)	Allergy N = 13,532 (%)	Anaphylaxis N = 548 (%)	Adjusted ROR of Allergy (95% CI)	Adjusted ROR of Anaphylaxis (95% CI)
**Sex**	Male	41,075 (43.66)	35,307 (44.14)	5507 (40.70)	261 (47.60)	1	1
	Female	52,832 (56.16)	44,531 (55.67)	8014 (59.22)	287 (52.40)	0.88 (0.84–0.91)	1.20 (1.01–1.42)
	Missing	166 (0.18)	155 (0.19)	11 (0.08)	-		
**Age, years**	0–1	2174 (2.31)	1816 (2.27)	358 (2.70)	-	1.19 (1.06–1.34)	-
	2–11	2954 (3.14)	2003 (2.50)	948 (7.00)	3 (0.55)	2.97 (2.74–3.22)	0.21 (0.07–0.66)
	12–17	1616 (1.72)	1222 (1.53)	386 (2.90)	8 (1.46)	1.88 (1.68–2.12)	0.97 (0.48–1.95)
	18–65	42,303 (44.97)	33,289 (41.61)	8657 (64.00)	357 (65.14)	2.48 (2.39–2.58)	2.63 (2.21–3.14)
	>65	44,866 (47.69)	41,518 (51.90)	3170 (23.40)	178 (32.48)	0.28 (0.27–0.29)	0.44 (0.37–0.53)
	Missing	160 (0.17)	145 (0.19)	13 (0.09)	2 (0.37)		
	Mean (±SD)	58.50 ± 23.90	60.59 ± 23.46	46.28 ± 22.98	55.70 ± 17.67		
**Ethnicity**	Caucasian	75,668 (80.44)	64,158 (80.20)	11,059 (81.72)	451 (82.30)	1.12 (1.07–1.78)	1.16 (0.93–1.45)
	Other	2953 (3.14)	2330 (2.92)	603 (4.46)	20 (3.65)	-	-
	Missing	15,452 (16.42)	13,505 (16.88)	1870 (13.82)	77 (14.05)		
**Triage codes**	Red	2672 (2.84)	2251 (2.81)	292 (2.16)	129 (23.54)	0.86 (0.76–0.97)	10.68 (8.69–13.13)
	Yellow	18,596 (19.77)	16,037 (20.05)	2376 (17.56)	183 (33.39)	0.92 (0.88–0.97)	2.00 (1.68–2.39)
	Green	36,045 (38.32)	29,748 (37.19)	6235 (46.07)	62 (11.31)	1.29 (1.24–1.34)	0.22 (0.17–0.28)
	White	3436 (3.65)	2719 (3.40)	717 (5.30)	-	1.25 (1.15–1.37)	-
	Missing	33,324 (35.42)	29,238 (36.55)	3912 (28.91)	174 (31.76)		
**Hospitalization**	Yes	26,644 (28.32)	24,168 (30.20)	2105 (15.60)	371 (67.70)	0.53 (0.50–0.55)	5.62 (4.66–6.79)
	No	67,429 (71.68)	55,825 (69.80)	11,427 (84.40)	177 (32.30)	-	-

CI: confidence interval; IQR: interquartile range; ROR: reporting odds ratio; SD: standard deviation. Logistic regression analyses were used to estimate the reporting odds ratios (RORs) with 95% confidence intervals (CIs) of drug-related allergies and anaphylaxis adjusting for sex, age classes, and ethnicity.

**Table 2 jcm-11-02811-t002:** Variation of adrenaline use in the management of drug-related anaphylaxis events over the study period.

Year	Allergy Events N = 13,532	Anaphylaxis Events N = 548	Adrenaline Use N = 323 (% Row)
**2012**	1610	59	37 (62.71)
**2013**	2552	108	61 (56.48)
**2014**	2737	101	50 (49.50)
**2015**	1609	52	36 (69.23)
**2016**	804	36	19 (52.78)
**2017**	1206	65	41 (63.08)
**2018**	1838	64	45 (70.31)
**2019**	1176	63	34 (53.97)

The percentage of adrenaline use refers only to anaphylaxis cases. Overall, 58.94% of anaphylaxis (323 out of 548 cases) reported adrenaline use during ED management.

**Table 3 jcm-11-02811-t003:** Suspected drug classes associated with anaphylaxis.

	Anaphylaxis N = 608 (%)	Non-Allergy N = 104,366 (%)	Unadjusted ROR (95% CI)	Adjusted ROR (95% CI)
**Antibacterials**	327 (53.78)	10,744 (10.29)	9.99 (8.51–11.75)	8.75 (7.47–10.25)
Penicillins	218 (66.67)	5503 (51.22)	9.91 (8.37–11.73)	8.98 (7.57–10.65)
Cephalosporins	69 (21.10)	1291 (12.02)	10.09 (7.80–13.05)	10.75 (8.32–13.88)
Fluoroquinolones	28 (8.56)	1881 (17.51)	2.60 (1.77–3.80)	2.30 (1.57–3.38)
Macrolides	6 (1.83)	1155 (10.75)	0.88 (0.39–1.97)	0.75 (0.34–1.68)
Glycopeptides	5 (1.53)	347 (3.23)	2.45 (1.01–5.96)	2.28 (0.94–5.53)
Sulfamet./Trimetop.	1 (0.31)	337 (3.14)	0.50 (0.07–3.58)	0.42 (0.06–2.99)
**NSAIDs**	78 (12.83)	4980 (4.77)	2.88 (2.27–3.66)	2.18 (1.71–2.78)
Diclofenac	32 (41.03)	1058 (21.24)	5.33 (3.71–7.65)	4.45 (3.11–6.39)
Ketoprofen	24 (30.77)	1727 (34.68)	2.40 (1.59–3.62)	1.65 (1.08–2.52)
Ketorolac	5 (6.41)	291 (5.84)	2.91 (1.20–7.08)	2.26 (0.93–5.49)
Flurbiprofen	4 (5.13)	101 (2.03)	6.72 (2.47–18.31)	5.21 (1.91–14.25)
Indomethacin	3 (3.85)	191 (3.84)	2.66 (0.85–8.34)	2.09 (0.66–6.60)
Nimesulide	2 (2.56)	581 (11.67)	0.58 (0.14–2.33)	0.43 (0.11–1.74)
Etoricoxib	2 (2.56)	234 (4.70)	1.44 (0.36–5.82)	1.30 (0.32–5.26)
Dexibuprofen	2 (2.56)	39 (0.78)	8.68 (2.09–36.02)	7.88 (1.90–32.68)
**Radiology contrast agents**	42 (6.92)	554 (0.53)	13.73 (9.92–19.00)	11.52 (8.33–15.92)
Iomeprol	16 (38.10)	201 (36.28)	13.83 (8.26–23.17)	11.52 (6.89–19.25)
Iopromide	12 (28.57)	88 (15.88)	25.56 (12.81–43.33)	20.26 (11.04–37.19)
Iobitridol	4 (9.52)	40 (7.22)	17.06 (6.08–47.84)	14.96 (5.41–41.41)
Iodixanol	3 (7.14)	52 (9.39)	9.82 (3.06–31.54)	8.62 (2.70–27.53)
Iopamidol	3 (7.14)	36 (6.50)	14.91 (4.36–46.22)	13.40 (4.05–44.40)
Other contrast agents	4 (9.52)	68 (12.27)	10.03 (3.65–27.59)	7.44 (2.71–20.47)
**Analgesic drugs**	32 (5.26)	11,131 (10.67)	0.46 (0.33–0.66)	0.47 (0.33–0.67)
Paracetamol *	15 (46.88)	2152 (19.33)	1.19 (0.71–1.98)	1.02 (0.61–1.71)
Acetylsalicylic acid	11 (34.38)	5589 (50.21)	0.32 (0.18–0.58)	0.38 (0.21–0.68)
Tramadol	2 (6.25)	859 (7.72)	0.39 (0.10–1.58)	0.34 (0.09–1.39)
Pethidine	2 (6.25)	12 (0.11)	28.34 (6.33–126.95)	22.99 (4.92–107.37)
Other analgesics	2 (6.25)	254 (2.28)	1.34 (0.33–5.38)	1.44 (0.36–5.86)
**Antineoplastic drugs**	23 (3.78)	7887 (7.56)	0.47 (0.31–0.72)	0.41 (0.27–0.63)
Paclitaxel	7 (30.43)	645 (8.18)	1.85 (0.87–3.91)	1.56 (0.73–3.30)
Oxaliplatin	5 (21.74)	486 (6.16)	1.75 (0.72–4.24)	1.53 (0.63–3.72)
Cetuximab	2 (8.70)	101 (1.28)	3.36 (0.83–13.67)	3.35 (0.83–13.54)
Trastuzumab	2 (8.70)	166 (2.10)	2.05 (0.51–8.27)	1.78 (0.44–7.15)
Rituximab	2 (8.70)	546 (6.92)	0.62 (0.15–2.49)	0.59 (0.14–2.37)
Other antineoplas. drugs	5 (21.74)	1226 (15.54)	0.69 (0.28–1.66)	0.59 (0.25–1.43)

CI: confidence interval; NSAID: non-steroidal anti-inflammatory drug; ROR: reporting odds ratio. * Alone or in combinations. The total number of suspected drugs involved in anaphylaxis and non-allergy events is bigger than the number of cases because more than one suspected drug can be reported in a pharmacovigilance report form. Logistic regression analyses were used to estimate the reporting odds ratios (RORs) with 95% confidence intervals (CIs) of drug-related allergies and anaphylaxis adjusting for sex, age classes, and ethnicity.

**Table 4 jcm-11-02811-t004:** Characteristics of studies included in the systematic review.

Author, Year	Country	Period of Observation and Study Design	Participating Centres	Patients’ Selection	Age and Sex	Total of Patients	ED Visits for Allergy or Anaphylaxis	Drug-Related Allergy or Anaphylaxis	Hospitalization for Drug-Related Allergy or Anaphylaxis	Causative Drug Classes
**Asai, 2014 [21]**	Canada	2011–2012 Retrospective single centre study	Adult tertiary care ED	Diagnosis of anaphylaxis or allergic reactions (ICD-10 codes)	Median (IQR): 31.5 (26.4–44.0) years Females 66.3%	37,730	98	18	NR	Amoxicillin 16.7%
**Banerji, 2014 [22]**	USA	2006–2008 Retrospective database analysis	Truven Health MarketScan Commercial and Medicare Supplemental Databases (Truven, Ann Arbor, Mich)	Diagnosis of anaphylaxis (ICD9-CM codes)	Mean ± SD 48 ± 19 years Females 71%	716	716	716	205	NR
**Bellou, 2003 [52]**	France	1 year (1998) Retrospective single centre study	General hospital ED	Cases of suspected allergic reaction	Mean ± SD 55 ± 18.5 years Females 51%	324	324	25	Overall, 90	Beta-lactams 28% Macrolides 20% NSAIDs 52%
**Bielen, 2019 [53]**	Croatia	2012–2015 Retrospective single centre study	Tertiary care university hospital ED	Cases of hypersensitivity (SMQ)	<29 years: 8005 30–39 years: 7875 40–49 years: 8095 50–64 years; 17611 65–74 years: 13414 >75 years: 16982 Females 54.6%	71,982	3039	627	38	Antibiotics 44.7% Analgesics and NSAIDs 18.7%
**Budnitz, 2005 [23]**	USA	3 months Retrospective multicentre study; database analysis	NEISS-CADES database	Cases of ADE	<2 years: 56; 2–9 years: 62; 10–19 years: 44; 20–29: 66; 30–39 years: 59; 40–49 years: 84; 50–59 years: 65; 60–69 years: 57; 70–79: 58; ≥80 years: 47 Females 63.9%	598	155	155	4	Antibiotics 42.9% Non-opioid analgesics 29.3% Cardiovascular agents 24%
**Budnitz, 2006 [24]**	USA	2004–2005 Retrospective multicentre study; database analysis	NEISS-CADES database	Cases of ADE	0–4 years: 104,185; 5–17 years: 225,082; 18–44 years: 362,044; 45–64: 147,178; ≥65 years: 83,549 Females 44.7%	701,547 estimated annual ED visits	235,202 estimated annual ED visits	235,202 estimated annual ED visits	13,232 estimated annual ED visits	NR
**Budnitz, 2011 [25]**	USA	2007–2009 Retrospective multicentre study; database analysis	NEISS-CADES database	Cases of ADE	65–69 years: 2470; 70–74 years: 1840; 75–79 years: 2629; 80–84 years: 2476; ≥85 years: 2621 Females 59%	265,802 estimated annual ED visits	39,455 estimated annual ED visits	39,455 estimated annual ED visits	5617 estimated annual hospitalization	Cardiovascular agents Antibiotics
**Cho, 2019 [42]**	Korea	2012–2016 Cross-sectional multicentre study	7 community hospitals EDs	Cases of anaphylaxis (ICD-10 codes)	Mean ± SD 51.5 ± 16.0 Females 34.7%	325,857	1021	135	NR	NSAIDs 28.1% Antibiotics 15.6% Antibiotics and NSAIDs 3.7% Radiocontrast media 2.2%
**Cianferoni, 2001 [54]**	Italy	1985–1996 Retrospective chart review	University hospital ED	Diagnosis of acute anaphylaxis	Mean ± SD 42 ± 18 Females 45%	113	113	52	NR	Antibiotics 48% NSAIDs 35%
**Cohen, 2008 [26]**	USA	2004–2005 Retrospective multicentre study; database analysis	NEISS-CADES database	Cases of ADE	<1 year: 386; 1–4 years: 703; 5–8 years: 302; 9–12 years: 216; 13–18 years: 475 Females NR	6681	2802	2802	Overall, 5.1	Antibiotics 60.8% Analgesics 9.2% Multiple agents 6.7% Respiratory medications 5.9% Psychotropic medications 2.2%
**Cohen, 2018 [43]**	Israel	2013–2016 Retrospective single centre study	Paediatric hospital ED	Cases of allergic reactions or anaphylaxis (Anaphylaxis Criteria, Sampson et al.)	Mean 6.8 years (range 0–16 years) Females 34.7%	113,067	428	10	8 (1 of which in ICU)	NR
**Dennehy, 1996 [27]**	USA	30 days (1994) Single centre study	General hospital ED	Cases of drug-related illness	Mean ± SD 41.7 ± 22.5 years Females 50%	50	7	7	Overall, 8	NR
**Gabrielli, 2018 [28]**	Canada	2012–2016 Retrospective/prospective multicentre study	3 paediatric hospital and 1 general hospital EDs	Cases of anaphylaxis (diagnosis at ED presentation or ICD codes)	Median 49.4 (IQR 40.1–62.9) adults; median 8.00 (IQR 3.79–15.36) children Females: 71.9% adults; 47.1% children	884,000	1913	115 (64 adults; 51 children)	Admitted (5/51 = 9.8% children, 3/64 = 4.7% adults) Admitted ICU (1/51 = 2.0% children, 1/64 = 1.6% adults) Admitted hospital ward (4/51 = 7.8% children, 2/64 = 3.1% adults)	Beta-lactams (28.1% adults, 31.4% children) Quinolones (20.3% adults, 2% children) Other antibiotics (6.3% adults, 0% children) NSAIDs (20.3% adults, 21.6% children) Radiocontrast media (3.1% adults, 3.9% children)
**Goh, 2018 [44]**	Singapore	2014–2015 Prospective multicentre study	3 general hospital EDs	Cases of anaphylaxis (ICD-9 codes)	Median 23 years (range 3 months to 88 years and 9 months) Females 49.1%	7373	426	85 (66 adults; 19 children)	3	NSAIDs (24.2% adults, 52.6% children) Antibiotics (21.2% adults, 5.3% children) Paracetamol (3.0% adults, 10.5% children)
**Grunau, 2015 [29]**	USA and Canada	2007–2012 Retrospective multicentre cohort study	2 teaching hospital EDs	Diagnosis of allergic reaction	Median (IQR): 34 (27–47) years patients treated with steroids; 35 (26–49) years patients treated without steroids Females 60.9%	2701	2701	702	11	Anti-infective agents 48.9% Nervous system agents 10.3% (analgesics 2%) Radiocontrast media 3.7% NSAIDs 2.4%
**Hall, 2020 [57]**	Australia	2010–2015 Retrospective multicentre cohort study	5 university tertiary hospital EDs	Cases of antimicrobial anaphylaxis (ICD-10 codes)	Median 51 years (IQR 36–67) Females 61%	293	185	185	7 ICU admission	Overall (out of 185) Penicillins 39.9% Cephalosporins 35.1% Amino-penicillins 18.5% Amino-cephalosporins 17.0%
**Hampton, 2014 [30]**	USA	2009–2011 Retrospective multicentre study; database analysis	Administrative database 63 centres	Cases of psychiatric medication-related ADE	19–44 years: 49.4 (46.5–52.4) 45–64 years: 33.3 (30.7–35.9) ≥65 years: 17.3 (14.7–19.8) Females 61.9%	89,094 estimated annual ED visits	11,493 estimated annual ED visits	11,493 estimated annual ED visits	Overall, 17,188 estimated annual hospitalization	Zolpidem Quetiapine Alprazolam Lorazepam Haloperidol Clonazepam Trazodone Citalopram Lithium Risperidone
**Han, 2018 [45]**	Korea	2009–2014 Retrospective cohort study; database analysis	National insurance claim database of the Health Insurance Review and Assessment (HIRA)	Cases of drug hypersensitivity reactions (ICD-10 codes)	88,003 ≤19 years 169,103 20–44 years 180,535 45–64 years 97,408 ≥65 years Females 57.5%	535,049	3984 (T88.6 code)	3984 (T88.6 code)	184 (T88.6 code)	NR
**Harduar-Morano, 2011 [31]**	USA	2005–2006 Retrospective multicentre study	General hospital EDs	Diagnosis of anaphylaxis (ICD9-CM codes)	Mean ± SD 38.7 ± 21.46 Females 57%	2751	2751	228	54	NR
**Hitti, 2015 [46]**	Lebanon	July–December 2009 Retrospective single centre study	Tertiary care centre ED	Cases of acute allergic reaction (ICD-9 codes)	Mean ± SD 31.8 ± 19.2 years Females 42%	293	245	58	Overall, 1 patient was hospitalized	Antibiotics 8.2% NSAIDs 4.9%
**Hsin, 2011 [47]**	India	2000–2010 Retrospective single centre cohort study	General hospital ED	Diagnosis of anaphylaxis (ICD9 codes)	Mean age overall 43.3 years Female 47%	201	86	Overall, 161	NR	NSAIDs Antibiotics Chemotherapy Anti-epileptics Contrast media Immunotherapy Biologics H1N1 Vaccine Anaesthesia
**Huang, 2012 [32]**	USA	2004–2008 Retrospective single centre study	Paediatric hospital ED	Cases of anaphylaxis	Median (IQR) 8 (4 months–18 years) years Females 49%	192 (20 had multiple reactions)	192	19	Overall, 28	NR
**Jones, 2013 [33]**	USA	2004–2013 Retrospective multicentre study; database analysis	NEISS-CADES database	Cases of fluoroquinolone-associated hypersensitivity ADEs	Mean age overall 48.22 years Females 73.7%	102,536	1659	1422	96	Ciprofloxacin Levofloxacin Moxifloxacin Gemifloxacin Ofloxacin
**Kim MY, 2018 (a) [48]**	Korea	2011–2013 Retrospective multicentre study	2 tertiary hospitals and 1 secondary hospital EDs	Cases of anaphylaxis (ICD codes)	Mean ± SD 46 ± 17.1 Females 55.2%	194	194	151	NR	Antibiotics Acetylsalicylic acid Radiocontrast media NSAIDs
**Kim MY, 2018 (b) [49]**	South Korea	2003–2016 Retrospective single centre study	Tertiary university hospital ED	Cases of anaphylaxis (Korean Standard Classification of Disease)	Mean ± SD 41.1 ± 23.4 Females 48.2%	199	199	72	13	Overall (out of 199) Antibiotics 40.2% NSAIDs 33.3% Radiocontrast media 11.1%
**Ko, 2015 [50]**	Korea	2007–2014 Single centre study	Tertiary teaching hospital ED	Cases of anaphylaxis (Skin or mucosal tissue involvement; Respiratory compromise; Systolic blood pressure <90 mmHg or syncope; Gastrointestinal symptoms)	Mean ± SD 48.4 ± 15.7 years Females 54.9%	655	415	187	Overall, 3 patients were hospitalized	Radiocontrast media 70 NSAIDs 39 Cephalosporins 34 Anticancer agents 16
**Losappio, 2014 [55]**	Italy	2011 Retrospective single centre study	General hospital ED	Cases of allergic urticaria (ICD-9 codes)	Mean 35.4 years (range 0–90 years) Females 49.2%	44,112	459	92 (79 adults; 13 children)	NR	NSAIDs Beta-lactams
**Lovegrove, 2011 [34]**	USA	2006–2009 Retrospective multicentre study	Drug Abuse Warning Network (DAWN), 250 non-federal, short-stay general hospitals	Cases of antivirals-related ADE	<6 years: 139; 6–11 years: 103; 12–17 years: 58; 18–44 years: 332; 45–64 years: 161; ≥65 years: 89 Female 59.3%	879	274	274	Overall, 125	Amantadine Rimantadine Oseltamivir Zanamivir
**Lovegrove, 2019 [35]**	USA	2011–2015 Retrospective multicentre study; database analysis	NEISS-CADES database	Cases of antibiotics-related ADE in children (MedDRA)	2870 < 1–2 years 743 3–4 years 1187 5–9 years 1742 10–19 years Females 52.1%	6542	5763	5763	Overall, 265	Overall (out of 6542) Penicillins 59.7% Cephalosporins 11.2% Sulfonamides 9.5%
**Motosue, 2018 (a) [36]**	USA	2005–2014 Prospective observational study; database analysis	OLDW administrative database	Cases of anaphylaxis (ICD-9 codes)	Median 36 years (interquartile range 17–52) Females 57.5%	56,212	56,212	6720	Inpatient 717 and ICU 409	NR
**Motosue, 2018 (b) [37]**	USA	2008–2012 Retrospective study; database analysis	Administrative claims database (OptumLabs Data Warehouse)	Cases of anaphylaxis and anaphylactic shock (ICD-9 codes)	Median 42 years (range 1–87 years) Females 58.3%	7367	7367	1076	Overall, 532 ICU admission	NR
**Quiralte, 1997 [56]**	Spain	1992–1995 Prospective single centre study	University hospital ED	Cases of NSAIDs-related anaphylaxis	Mean ± SD 35.7 ± 13.9 Females 71%	21	21	21	NR	Dipyrone 57.1% Propyphenazone 14.2% Acetic derivatives (diclofenac and indomethacin) 14.2%
**Rangkakulnuwat, 2020 [51]**	Thailand	2007–2016 Retrospective single centre study	University hospital ED	Cases of anaphylaxis (ICD-10 codes)	Median 24.0 years (IQR 19.0–43.0) Females 57.2%	10,848,695	441	79	NR	NSAIDs 7.4% Antimicrobial Agents 4.0% Radiocontrast media 0.9%
**Russell, 2010 [38]**	USA	2002–2006 Retrospective single centre cross-sectional study	Tertiary care paediatric hospital ED	Diagnosis of anaphylaxis (ICD-9 codes)	Mean ± SD 9.49 ± 5.56 Females 36%	103	103	15	4	Antibiotics Intravenous contrast
**Schneitman Mcintire, 1996 [39]**	USA	1992–1993 Retrospective single centre study	General hospital ED	Patients who experienced medication misadventures	15–44 years 38% 65 years or older 33% Females 62%	62,216	221	204	7	Trimetoprim sulfametoxazol 34% Amoxicillin 21% Ibuprofen 5.4%
**Shehab, 2008 [40]**	USA	2004–2005 Retrospective multicentre study; database analysis	NEISS-CADES database	Cases of antibiotics-related ADE	<1 years: 545; 1–4 years: 976; 5–14 years: 656; 15–44 years: 2577; 45–64 years: 1143; 65–79 years: 507; ≥80 years: 210 Females 64.4%	142,505 estimated annual ED visits	112,116 estimated annual ED visits	112,116 estimated annual ED visits	8738 estimated annual hospitalization	Penicillins 36.9% Cephalosporins 12.2% Fluoroquinolones 13.5% Sulfonamide trimethoprim 11.8% Macrolides and Ketolides 6.9% Tetracyclines 3.1% Vancomycin Linezolid 0.8%
**Willy, 2009 [41]**	USA	2004–2005 Retrospective multicentre study; database analysis	NEISS-CADES database	Cases of analgesics-related ADE	0–9 years: 32,222; 10–19 years: 17,012; 20–29 years: 28,298; 30–39 years: 23,165; 40–49 years: 22,706; 50–59 years: 18,767; 60–69 years: 14,590; 70–79 years: 15,030; 80–89 years: 14,933; ≥90 years: 1998 Females 57%	188,721	58,101	58,101	Overall, 22,646	Acetaminophen Non-narcotic-acetaminophen combination Narcotic- acetaminophen combination Acetylsalicylic acid Ibuprofen Naproxen

ADE: adverse drug event; ED: emergency department; ICD-CM: International Classification of Diseases-Clinical Modification; ICU: intensive care unit; IQR: interquartile range; MedDRA: Medical Dictionary for Regulatory Activities Terminology; NR: not reported; NSAIDs: non-anti-inflammatory drugs; SD: standard deviation; SMQ: standardized MedDRA query; T88.6 code: Anaphylactic shock due to adverse effect of correct drug or medication properly administered.

## Data Availability

Data that support the findings of this study are available upon reasonable request from the corresponding author, N.L.

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
