# Peer review of "Drug-Related Hypersensitivity Reactions Leading to Emergency Department: Original Data and Systematic Review"

_jcm, 2022, doi:10.3390/jcm11102811_

Round 1
Reviewer 1 Report
extensive data collection and review did by the research team.
need to clarify systemic reviews methodology- line 136 mentioned that systemic reviews and meta-analysis were considered
but in 144 it is mentioned that meta-analysis was excluded
table 1 needs to revise with the correct numbers for overall cases. there are some irregularities in overall cases.
if there are any missing data, mention how many cases information are missing in each category.
Author Response
Reviewer #1:
Extensive data collection and review did by the research team. Need to clarify systemic reviews methodology - line 136 mentioned that systemic reviews and meta-analysis were considered but in 144 it is mentioned that meta-analysis was excluded.
Dear Reviewer 1, we apologize for the misunderstanding, but, in line 136, where we mentioned “meta-analysis”, we only referred to the PRISMA guidelines for systematic reviews and meta-analysis conduction and to its specific reference. The aims of our research did not comprehend the conduction of a meta-analysis, as already stated throughout the manuscript, in particular: “The aim of the present study was to describe pharmacological characteristics of drug-related allergies and anaphylaxis leading to ED in Italy, estimating their risk considering subjects’ demographic and clinical characteristics, and the most frequently reported suspected drug classes. Furthermore, to complete the evidence obtained with our post-hoc analysis, a systematic review of observational studies on drug-related hypersensitivity ADEs leading to ED visit and/or hospitalization was performed.”. Moreover, in line 144, following the PRISMA guidelines, we correctly reported the type of studies that were excluded from the systematic review, including meta-analysis.
Table 1 needs to revise with the correct numbers for overall cases. there are some irregularities in overall cases. If there are any missing data, mention how many cases information are missing in each category.
Dear Reviewer 1, we thank for your suggestion and we confirm the correction of Table 1, where we also reported missing data. Moreover, we added this important issue in the Limitations section, as follows: “Our analysis is based on ADE reports that are affected by limits that include inaccurate and incomplete information on patients (i.e., sex, age, ethnicity, triage colour code), mainly related to lack of clinical data also in the ED electronic sources.”.
Reviewer 2 Report
The authors reported the data regarding to the drug-related hypersensitivity reactions leading to emergency departments in Italy. It was a multi-center study recruiting the database of patients from 2012 to 2019. The results were informative. I Have just a few suggestions as follow.
- The multivariate logistic regression showed a lower risk of drug-related allergy in females. This looks overstated. The result just showed among those patients visiting the emergency departments, females had a lower risk to be diagnosed as drug-related allergy. It might be resulted from obviously more females visiting the emergency departments than males in your database. The authors should carefully present and explain the results.
- Could the authors provide more details about the types of drug-related allergy? For examples, maculopapular eruption, urticaria, sick-sinus syndrome, vasculitis, and severe cutaneous adverse reactions (SCARs)...
- Few typos in the manuscript should be corrected.
Author Response
Reviewer #2:
The authors reported the data regarding to the drug-related hypersensitivity reactions leading to emergency departments in Italy. It was a multicenter study recruiting the database of patients from 2012 to 2019. The results were informative. I Have just a few suggestions as follow.
The multivariate logistic regression showed a lower risk of drug-related allergy in females. This looks overstated. The result just showed among those patients visiting the emergency departments, females had a lower risk to be diagnosed as drug-related allergy. It might be resulted from obviously more females visiting the emergency departments than males in your database. The authors should carefully present and explain the results.
We thank Reviewer 2 for this important consideration. In fact, although our post-hoc analysis showed that female was the sex most represented in allergy events, in our cohort (ED setting) women seem to be associated with a lower risk of drug-related allergy but with a higher risk of anaphylaxis, even if at the limit of statistical significance. Considering drug-related allergy, this evidence is comparable with real-world data coming from other observational studies, in which a higher risk of ADEs was usually observed in women. Nevertheless, there is no conclusive data that drug-related allergies are more common in females than in males. Following Reviewer 2’ suggestion, we stated these considerations in the first part of the Discussion section, as follows: “Although our post-hoc analysis showed that female was the sex most represented in allergy events, women seem to be associated with a lower risk of drug-related allergy but with a higher risk of anaphylaxis, even if at the limit of statistical significance. Considering drug-related allergy, this evidence is comparable with real-world data coming from observational studies, in which a higher risk of ADEs was usually observed in women [58, 59]. Nevertheless, there is no conclusive data that drug-related allergies are more common in females than in males [60].”.
Could the authors provide more details about the types of drug-related allergy? For examples, maculopapular eruption, urticaria, sick-sinus syndrome, vasculitis, and severe cutaneous adverse reactions (SCARs)...
We thank Reviewer 2 for this valuable comment. We added a briefly description of the most frequently reported ADEs within the Results section, as follows: “For all suspected drug classes, the most frequently reported drug-related hypersensitivity reactions affected the skin and subcutaneous tissue (data not shown). In particular, we observed several cases of urticaria, localised or general pruritus, erythema, and rash. These dermatological manifestations showed a different degree of seriousness among patients. Considering the most severe cases, these were represented by systemic reactions, including respiratory distress and anaphylactic shock.”.
Few typos in the manuscript should be corrected.
We confirm that we have corrected typos throughout the manuscript.

Round 2
Reviewer 2 Report
The manuscript has improved after revision. However, the limitations of the retrospective database make the results less convincing.